# Current Knowledge on Epizootic Haemorrhagic Disease in China

**DOI:** 10.3390/vaccines11061123

**Published:** 2023-06-20

**Authors:** Jige Xin, Jun Dong, Jing Li, Lingling Ye, Chong Zhang, Fuping Nie, Yeqing Gu, Xincheng Ji, Zhigang Song, Qianmin Luo, Jun Ai, Diangang Han

**Affiliations:** 1College of Veterinary Medicine, Yunnan Agricultural University, Kunming 650201, China; 2007032@ynau.edu.cn (J.X.); guyeqing446@outlook.com (Y.G.); 2Animal Quarantine Laboratory, Technology Center of Kunming Customs, Kunming 650200, China; km29643@sina.com (J.D.); lijing9401@126.com (J.L.); ye_ling08@sina.com (L.Y.); leofighting@163.com (C.Z.); 18288947175@163.com (Q.L.); 13769143048@163.com (J.A.); 3Animal and Plant Quarantine Laboratory, Technology Center of Chongqing Customs, Chongqing 400020, China; nie1626@163.com; 4Research Center for International Inspection and Quarantine Standard and Technical Regulation, General Administration of Customs, Beijing 100013, China; jixincheng@126.com (X.J.); songzhigang@hotmail.com (Z.S.)

**Keywords:** epizootic haemorrhagic disease (EHD), distribution, detection methods, China

## Abstract

Epizootic haemorrhagic disease (EHD) is an infectious, non-contagious viral disease of ruminants caused by epizootic haemorrhagic disease virus (EHDV) and is transmitted by insects of the genus *Culicoides*. In 2008, EHD was listed on the World Organization for Animal Health (WOAH) list of notifiable terrestrial and aquatic animal diseases. This article reviews the distribution of EHD in China and relevant studies and proposes several suggestions for the prevention and control of EHD. There have been reports of positivity for serum antibodies against EHDV-1, EHDV-2, EHDV-5, EHDV-6, EHDV-7, EHDV-8 and EHDV-10 in China. Strains of EHDV-1, -5, -6, -7, -8 and -10 have been isolated, among which the Seg-2, Seg-3 and Seg-6 sequences of serotypes -5, -6, -7 and -10 belong to the eastern topotype. The emergence of western topotype Seg-2 in EHDV-1 strains indicates that EHDV-1 strains in China are reassortant strains of the western and eastern topotypes. A novel serotype strain of EHDV named YNDH/V079/2018 was isolated in 2018. Chinese scholars have successfully expressed the EHDV VP7 protein and developed a variety of ELISA detection methods, including antigen capture ELISA and competitive ELISA. A variety of EHDV nucleic acid detection methods, including RT–PCR and qRT–PCR, have also been developed. LAMP and the liquid chip detection technique are also available. To prevent and control EHD, several suggestions for controlling EHD transmission have been proposed based on the actual situation in China, including controlling the number of *Culicoides*, reducing contact between *Culicoides* and hosts, continued monitoring of EHDV and *Culicoides* in different areas of China and further development and application of basic and pioneering research related to EHD prevention and control.

## 1. Introduction

Epizootic haemorrhagic disease (EHD) is an infectious, non-contagious viral disease of ruminants caused by epizootic haemorrhagic disease virus (EHDV) and is transmitted by insects of the genus *Culicoides*. It mainly affects a variety of domestic and wild ungulates, such as white-tailed deer and cattle. EHD can result in decreased milk production, mouth lesions and mortality in cattle and considerable mortality, chronic lesions and hoof abnormalities in white-tailed deer [1,2]. EHD has a negative impact not only on the local economies of regions and countries but also on the ecology and health of threatened or endangered wild ruminant populations [3,4]. In 2008, EHD was listed on the World Organization for Animal Health (WOAH) (https://www.woah.org/en/what-we-do/animal-health-and-welfare/animal-diseases/ accessed on 13 June 2023) list of notifiable terrestrial and aquatic animal diseases.

EHDV belongs to the genus Orbivirus within the family Reoviridae and is closely related to bluetongue virus (BTV) and African horse sickness virus. It has a 10-segmented double-stranded RNA (dsRNA) genome encoding seven structural proteins (VP1–VP7) and four non-structural proteins (NS1, NS2, NS3/NS3a and NS4). The protein VP2 of the outer core is the major determinant of serotype specificity, while the VP7 of the inner core possesses serogroup-specific antigens [5]. The inner core proteins (VP1, VP3, VP4 and VP6) exhibit high levels of sequence conservation and demonstrate strong topotype ability for isolates in eastern or western groups [6]. VP3, encoded by Seg-3, is a highly conserved major protein that constitutes the inner capsid of the virus. As the nucleic acid sequence of Seg-3 varies among strains isolated from different regions, the topotype of the virus can be determined based on this sequence. EHDV strains isolated worldwide are divided into eastern-type and western-type strains [7,8]. The sequence variation in Seg-3 in the east and west groups has reached 13.3%, indicating a significant divergence between the two populations [6]. To date, seven EHDV serotypes (EHDV-1, EHDV-2, EHDV-4~8) have been recognized, and two putative serotypes (EHDV-9~10) have been reported [5,9]. The early isolated EHDV-3 (NIG/1967-1 strain) was confirmed as EHDV-1 by means of Seg-2 gene sequencing. EHDV-9, which does not have a gene sequence in GenBank, was originally isolated from an alpaca in South Africa in 2013. EHDV-10 (ON-4/B/98 strain) was first isolated in an asymptomatic cow in Okinawa, Japan [10,11].

In America, EHDV was first isolated in 1955 following an outbreak in white-tailed deer in New Jersey. However, the virus may have been present in America long before that time, as anecdotal reports consistent with haemorrhagic disease in deer date back to the late 1800s [12]. At that time, the condition was described by hunters as black tongue, which was later recognized as EHD [4]. EHDV has been known to circulate outside of North America since shortly after its initial discovery. Globally, EHDV has been identified broadly across temperate and tropical regions of the Americas, Asia, Africa, Australia and the Middle East [13]. The prevalence of different serotypes of EHDV varies greatly in different regions of the world. For example, EHDV-1, EHDV-2 and EHDV-6 are common in North America and South America, whereas EHDV-1, EHDV-2, EHDV-7 and EHDV-10 are common in Japan [14]. Up to now, EHDV in North America has spread north from the United States to Alberta, Ontario, Canada; in South America, EHDV has spread to Ecuador, French Guadeloupe, Trinidad and Tobago, French Guiana and Brazil. In Africa, EHDV has spread to Reunion in France, Mauritania in West Africa and Egypt and Libya in northeastern North Africa. In Asia, EHDV has spread to Japan, Israel and China. In Oceania, EHDV is currently confined to Australia and has not spread [15]. In Europe, no cases of EHD have been reported before 2022, but it has been reported in neighboring Asian and African countries such as Tunisia, Turkey, Algeria and Israel. The first cases of EHD in Europe were detected in Italy and Spain in 2022, which are also the most recent cases reported by WOAH. The EHDV in Italy belongs to EHDV-8, while the EHDV in Spain has not yet been identified (https://wahis.woah.org/#/event-management accessed on 13 June 2023).

*Culicoides* midges are the only known arthropod vector for EHDV, with a key role in the life cycle and viral dissemination from infected to susceptible vertebrate hosts [4] (Figure 1). *Culicoides* are found all over the world except New Zealand and Antarctica. More than 1400 species of *Culicoides* are known, of which more than 30 can transmit EHDV [16]. The global distribution of EHDV depends on the arthropod vector’s geographical occurrence and distribution and susceptible vertebrate hosts. Overall, vector competence can vary for different EHDV strains within a *Culicoides* species, impacting the geographic distribution of EHDV serotypes [4]. Factors affecting the competence of *Culicoides* for viral dissemination include the virus serovar and arthropod vector genotype as well as environmental factors such as temperature and humidity [17,18].

As a transboundary disease, EHD is gradually spreading around the world along with global trends such as changing climate and ecological conditions, trade and human behaviour, causing huge economic losses and an increased the burden on public health in countries around the world. China has a vast territory and complex natural conditions, and many locations are suitable for the growth and reproduction of various *Culicoides*, which is conducive to transmission of EHDV. With the increase in communication between China and the rest of the world, EHD has also had a very adverse impact on China’s livestock industry.

## 2. Serological Investigation, Isolation and Identification of EHDV

China is a leading country in the world of cattle and sheep breeding, with 102.16 million cattle herds and 326.27 million sheep herds in 2022 (http://www.ce.cn/xwzx/gnsz/gdxw/202301/18/t20230118_38353331.shtml accessed on 13 June 2023). Cattle and sheep breeding plays an important role in China’s economy, but the prevalence of EHDV in China has been unclear for a long time. Serological investigation of EHDV is helpful to understand the epidemic status of EHDV in herds and to scientifically assess the future development trend of EHD and has important significance for formulation and improvement in the prevention of and control strategies for EHD. Since 2013, China has carried out serological investigation of EHDV across the country. The results show that animals seropositive for EHDV exist in both northern and southern China, with the rate of positivity gradually increasing from north to south. Moreover, there are multiple serotypes of the EHDV epidemic, and the serotypes of EHDV present in different regions are also different [19,20]. There have been reports of seropositive cases of EHDV-1, EHDV-2, EHDV-5, EHDV-6, EHDV-7, EHDV-8 and EHDV-10 in China, and there is a risk of an epidemic of EHD in China [8,21,22] (Figure 2). Yunnan Province is located on the southwest border of China in tropical and subtropical regions, bordering Myanmar, Laos and Vietnam. It has high annual rainfall, a long annual sunshine duration and high relative humidity and is suitable for the growth and reproduction of *Culicoides*. Yunnan Province is known as the kingdom of flora and fauna in China due to its biodiversity and is also a very representative area for EHD research. To understand the prevalence and distribution of EHD in Yunnan Province, Kou et al. [22] tested 1199 bovine serum samples from 11 countryside locations in Yunnan Province for EHDV antibody in 2019. At the same time, 18~50 positive sera were randomly selected for serotype identification. The average positive rate of EHDV was 81.8%. The positive rates of the EHDV-7, -10, -6 and -5 serotypes were high, at 25.2%, 23.2%, 22.2% and 15.4%, respectively, but the rates of the EHDV-8 and -2 serotypes were low, at 6.3% and 1.8%, respectively. EHDV-2 was detected only in Mengla County. Based on the distribution statistics of antibody detection results, it was found that the positive rate was higher on livestock farms, followed by the live livestock trading market, and was lowest in free-range households.

Isolation, identification and genetic analysis of EHDV strains provide a basis for aetiological diagnosis, epidemiology and pathogenicity of EHD and are of great significance for the scientific development of EHD prevention and control strategies. Chinese academics have isolated strains of EHDV-5, EHDV-6, EHDV-7, EHDV-8 and EHDV-10 in Yunnan, Guangdong and Guangxi, and the Seg-2, Seg-3 and Seg-6 sequences of EHDV-5, EHDV-6, EHDV-7 and EHDV-10 were confirmed to belong to the eastern topotype, which is closely related to strains isolated in Japan or Australia [3,8,10,20,21,23,24,25]. Li et al. [26] reported that phylogenetic trees based on the nucleotide sequence similarity of Seg-2 indicate that the seven strains of EHDV-1 isolated in Yunnan, Guangdong and Guangxi belon to the western topotype. However, phylogenetic trees based on the nucleotide sequence similarity of Seg-3 and Seg-6 indicate that the seven strains of EHDV-1 belong to the eastern topotype. The emergence of western topotype Seg-2 in EHDV-1 strains isolated in China indicates that western topotype strains may reassorted with local epidemic EHDV-1 strains after invading China, and a similar situation also existed in the BTV reassorted strain (e.g., for BTV-1(w), BTV-4(w), BTV-8(w) and BTV-16(e)) [27]. The study provides a warning that the invasion of western topotype strains into China could lead to severe economic losses in the animal industry and so must be prevented. In 2018, researchers from the Yunnan Animal Science and Veterinary Institute isolated a strain of EHDV named YNDH/V079/2018 from a sentinel calf in Mangshi County, Yunnan Province, China. Nucleotide sequencing and neutralization tests indicated that the virus belongs to a novel serotype of EHDV that had not been reported previously. In addition, the genetic and infective characteristics of the novel serotype EHDV were analysed [28,29] (Figure 3). The study provides a basis for further research into the epidemiological investigation and pathogenicity of this novel serotype EHDV strain.

## 3. Study on Detection Methods of EHDV

Studies on EHD detection methods are of great significance to carry out epidemiological investigations of EHDV, formulate prevention and control strategies and safeguard import and export trade. The VP7 protein is encoded by Seg-7, and its amino acid sequence is highly conserved among different serotypes of EHDV strains [30]. As a group-specific antigen of EHDV, the VP7 protein induces infected animals to produce group-specific antibodies against EHDV and is the most important candidate antigen for the development of diagnostic reagents for EHDV. Yang et al. [31] constructed the expression carrier pBAD-EHDV-VP7 and expressed it through a prokaryotic expression system to obtain the recombinant VP7 protein with high purity. The expressed VP7 protein reacts specifically with EHDV-positive sera. Li et al. [32] successfully expressed the EHDV VP7 protein in Escherichia coli. The recombinant protein has good reactivity and immunogenicity, and the rabbit polyclonal antibody against the EHDV VP7 protein obtained by immunizing rabbits has a titre of 1: 24,000, with good reactivity and specificity, which provides test materials for preparation of the EHDV diagnostic antigen and establishment of a serological detection method. Guo et al. [33] obtained two hybridoma cell lines (1A5 and 8H6) that stably secrete anti-EHDV monoclonal antibodies. The two monoclonal antibodies were identified with high stability and good specificity, which laid the foundation for development of rapid detection of antigen by ELISA. In research on ELISA detection methods, Chinese scholars have established antigen capture ELISA, competitive ELISA, double-antibody sandwich ELISA, indirect ELISA and blocking ELISA [34,35,36,37,38]. All established ELISA methods exhibit excellent specificity. For instance, the antigen capture ELISA exclusively detects antibodies to EHDV of distinct serotypes without any cross-reactivity with bluetongue virus (BTV), Akabane virus (AKAV) and Chuzan virus (CHUV). When detecting EHDV-positive sera of different serotypes using the established competitive ELISA method, all results were positive; however, detection outcomes for Akabane virus (AKAV), Chuzan virus (CHUV), peste des petits ruminants virus (PPRV), Foot-and-mouth disease virus (FMDV) and bovine ephemeral fever virus (BEFV) were negative. Different ELISA methods provide a rapid, sensitive, stable and effective method for EHDV detection and EHD epidemiological investigation. Dong et al. [35] developed a blocking ELISA detection method after optimizing the reaction conditions with inactivated EHDV as the acetylating antigen and a horseradish peroxidase (HRP)-conjugated goat anti-mouse IgG secondary antibody as the detection antibody. The specificity and sensitivity of the method were consistent with those of the ID Screen^®^ EHDV kit, and the coincidence rate was 97.00%.

PCR assays are widely used in pathogen detection because of their simple operation, high sensitivity and low requirements for sample quantity and quality. Yang used Seg-2, which determines the serotype of EHDV, as the detection target gene and designed seven pairs of specific primers to establish a RT–PCR detection method for identifying the serotype of EHDV [39]. The specific amplification primers and TaqMan probes were designed to establish a serotype-specific real-time fluorescence quantitative RT–PCR (qRT–PCR) detection method for EHDV [40]. The two detection methods can accurately identify different serotypes of EHDV strains, and there is no cross-reaction with bluetongue virus (BTV), Chuzan virus (CHUV) or Akabane virus (AKAV), with good specificity and high sensitivity; it can be used for rapid and accurate diagnosis of EHDV serotypes in animals. Conventional RT–PCR and qRT–PCR were established for Seg-2 of the novel EHDV strain (YNDH/V079/2018) [28,29], which showed good specificity and sensitivity. There was no cross-reaction with other serotypes of EHDV, BTV, CHUV or AKAV. The lower limit of viral nucleic acid detection was 10 copies/μL for qRT–PCR and 10^2^ copies/μL for RT–PCR. The two methods have been used to provide effective techniques for early diagnosis, epidemiological investigation and experimental study of the new serotype EHDV. Aiming at the conserved region of the VP7 gene of EHDV nucleic acid, Yang et al. [41] designed and synthesized a pair of specific primers and established a one-step RT–PCR detection method; Jiang et al. [42] established a fluorescence quantitative RT–PCR by designing primers and TaqMan fluorescent probes. The multiplex PCR method, which detects multiple pathogenic nucleic acids at the same time, can efficiently detect related pathogens and play an important role in epidemiological investigation. Yang et al. designed specific primers and TaqMan probes for BTV and EHDV according to the Seg-10 sequence of BTV and Seg-5 sequence of EHDV, respectively. After optimizing the reaction conditions, a duplex fluorescence quantitative RT–PCR method for simultaneous detection of two viruses was established [43]. The established dual fluorescence quantitative RT–PCR method has the advantages of being rapid, accurate, sensitive and stable. In addition, Yang et al. designed three pairs of specific primers and probes to establish a triple RT–qPCR detection method by selecting NS3 of BTV, NS1 of EHDV and VP7 of Palyam serogroup virus (PALV) as target genes [44]. The method can detect 24 BTV serotypes, 6 EHDV serotypes and 3 PALV serotypes with group specificity. The copy number detection limits of in vitro-transcribed ssRNAs were all as low as 10 μL^−1^, which is equivalent to those of a single-plex RT-qPCR assay for each virus. The established triplex RT-qPCR assay could effectively detect the positive blood samples collected from animals infected by BTV, EHDV and PALV.

RT-LAMP is a very suitable method for rapid detection in the field. The specific primers were designed based on the conserved region of the VP7 gene encoding inner capsid proteins of EHDV, and the RT-LAMP assay was developed through optimization of reaction conditions [45,46]. The developed RT-LAMP assay is capable of detecting four serotypes of EHDV RNA with no cross-reactivity to FMDV, BVDV, BTV or AHSV RNA. The detection limit of the RT-LAMP assay is 4.7 × 10^−8^ ng/L of RNA, indicating it is 1000 times more sensitive than conventional RT-PCR methods. The established EHDV RT-LAMP method boasts strong specificity, high sensitivity, rapid detection and high-throughput capabilities, which makes it very suitable for the field detection of EHDV infection. The liquid chip detection technique is a rapid and high-throughput technology for identifying the presence of viruses that can significantly reduce costs and work intensity. Zhan et al. [47] developed an EHDV-specific probe targeting the VP7 gene and labelled it with biotin. After coupling fluorescence-encoded microspheres and hybridizing them with PCR products of the viral VP7 gene, they established a rapid and high-throughput liquid chip detection method for EHDV. The method exhibits excellent specificity without cross-reactivity to BTV, VSV, WNV or AKV. Sensitivity testing demonstrated that the limit of detection for EHDV could reach 100 TCID_50_. The development of the Liquichip detection technique provides a foundation for further research on rapid and high-throughput detection of EHDV.

## 4. Recommendations for EHD Prevention and Control

In recent years, the geographical distribution of different serotypes of EHDV and their transmission vectors has expanded globally, and the severity and frequency of infection have gradually increased, which seriously endangers the ruminant breeding industry. Although there has been no regional outbreak of EHD in China, seropositive cases of EHD have been found in Guangdong, Guangxi, Inner Mongolia, Yunnan and other provinces, posing a risk for the disease. To prevent and control EHD, we can start from the following aspects.

### 4.1. Control the Number of Culicoides and Reduce Contact between Culicoides and Hosts

In areas where *Culicoides* exists, efforts should be made to prevent suitable conditions for *Culicoides*, such as low-lying undergrowth and natural grassland. *Culicoides* should be eliminated. The breeding density of EHD hosts, such as cattle and sheep, should be reduced, and efforts should be made to isolate *Culicoides*, such as installing physical isolation devices in enclosures. High-density host rearing can potentially increase contact opportunities between hosts and vectors and may consequently act as hotspots for EHD transmission. Additionally, increased vector–host interactions may result in more feeding by infected vectors within a single gonotrophic cycle. If an infectious *Culicoides* is interrupted while feeding on one host, the insect may move between hosts to acquire a full blood meal, potentially enhancing pathogen transmission to multiple hosts concurrently [48]. As *Culicoides* have crepuscular feeding activity [49], reducing host outdoor activities at dusk may help reduce individual animal contact with infectious vectors during *Culicoides* peak activity time. Female *Culicoides* require a blood meal for maturation of the ovaries and egg production and are known to attack vertebrates [4]. Parous female *Culicoides* are most likely to be infectious vectors of EHDV due to their physiological state. Therefore, reducing contact between the host and female parous *Culicoides* can reduce the infection rate of the host.

### 4.2. Long-Term Surveillance of EHDV and Culicoides in Different Areas and Risk Assessment

As domestic and international communications increase, the risk of an accidental incursion of EHDV is likely to increase in previously EHDV-free areas. Global warming will also certainly affect the distribution and active period of vectors. To prevent damage to animal husbandry, continuous surveillance of EHDV and *Culicoides* should be strengthened. Most arboviruses generally result in mild morbidity, asymptomatic/mild illnesses or no apparent acute phase in domestic ruminants, and thus, they silently circulate in endemic regions. Active surveillance, such as sentinel and vector surveillance systems, is therefore needed for efficient detection of arbovirus circulation in the field prior to disease outbreaks [50]. Cattle are commonly used as sentinels for EHD surveillance, but isolation from sentinel cattle is highly limited due to the short viraemia of bovine arboviruses [51]. It would be necessary to modify the frequency and period of blood sampling for future monitoring in this area. The status of infections in domestic ruminants in many countries bordering China, such as Myanmar, Laos, Mongolia and North Korea, remains unclear. Strengthening EHDV surveillance in border areas and cracking down on animal smuggling and quarantine evasion can reduce the risk of EHDV being introduced into China from abroad and support planning of preventive and control strategies for EHDV infection in other parts of China. It is necessary to continuously monitor the virulence and pathogenicity of EHDV and to further isolate, identify and genetically analyse EHDV, which will provide an important basis for prevention and control of EHD.

### 4.3. Basic Research and Applications Related to EHD Prevention and Control Should Be Further Carried Out

EHDV is transmitted between ruminant hosts by *Culicoides*. Further studies on the systematics, biology and vector competence of *Culicoides* will help in developing better surveillance and preventive measures for EHDV infection. The development of sensitive, specific, convenient and rapid EHDV detection technologies can improve the level of detection, which is crucial for the identification of the invasion and transmission of EHDV. When next-generation sequencing (NGS) technologies are used to monitor EHDV, the results will contribute to the development of detection systems suitable for EHDV and knowledge of arbovirus aetiology. Similarly, accurate risk assessment models, vector control technology, vaccine research and development are conducive to the prevention and control of EHD. Vaccines have been used to control EHD in the USA and Japan. In the USA, autogenous inactivated vaccines were developed for captive wildlife deer farmers using EHDV isolates from ill or dead animals in affected premises. Their use must be approved by government authorities. In Japan, the live attenuated vaccine was developed based on the Ibaraki-2 strain, while the inactivated vaccine was developed from bovine ephemeral fever and Ibaraki viruses grown in cell cultures and then inactivated using formalin. The administration of both vaccines is based on the current epidemiological situation in Japan and is voluntary. In areas where outbreaks of EHD occur, especially with the highly pathogenic EHDV strains, government-authorized vaccines can be employed for preventive purposes, as is practiced in the United States and Japan. The vaccination strategies should be tailored to the specific characteristics of each type of vaccine. The live attenuated vaccine has to be administered subcutaneously once during the low vector season. Inactivated vaccines are safer, but require multiple doses to be effective, and a yearly booster is recommended (https://www.woah.org/en/disease/epizootic-haemorrhagic-disease/ accessed on 13 June 2023). Moreover, efforts should be directed towards the development of an affordable and completely safe vaccine that can provide quick, lifelong and broad protection against all susceptible ruminant species. EHDV shares many morphological and structural characteristics with bluetongue virus (BTV). In susceptible species, EHDV may cause a disease with clinical manifestations similar to BTV infection. The development of an EHDV vaccine could model the development of a BTV vaccine. Experimental BT vaccines, including protein vaccines, viral vector vaccines and replicating vaccines, have been developed and extensively researched [52].

## 5. Conclusions

In conclusion, EHD seriously affects the health of ruminants and the healthy development of breeding industry. Positive cases of EHDV antibody have been reported in China, and multiple serotypes of EHDV have also been isolated. In view of the epidemic situation of EHD in the world, especially in China and neighbouring countries, comprehensive prevention and control measures should be taken to prevent a large-scale outbreak of EHD in China, such as strengthening continuous EHD surveillance, in-depth EHDV-related research, research and development of effective EHD vaccine and isolation of Culicoides and susceptible hosts.

## Figures and Tables

**Figure 1 vaccines-11-01123-f001:**
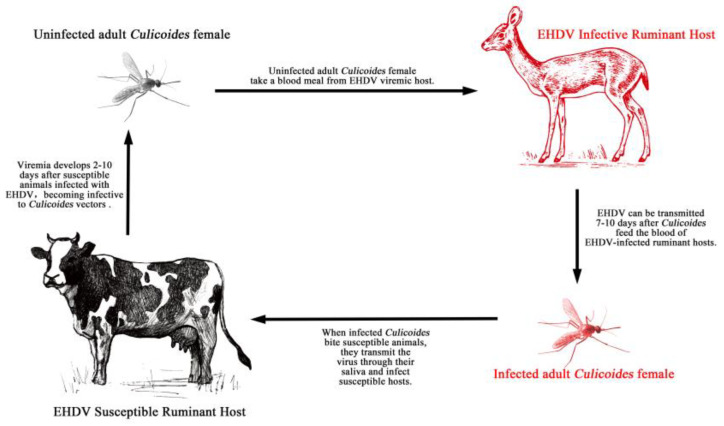
Life cycle of epizootic haemorrhagic disease.

**Figure 2 vaccines-11-01123-f002:**
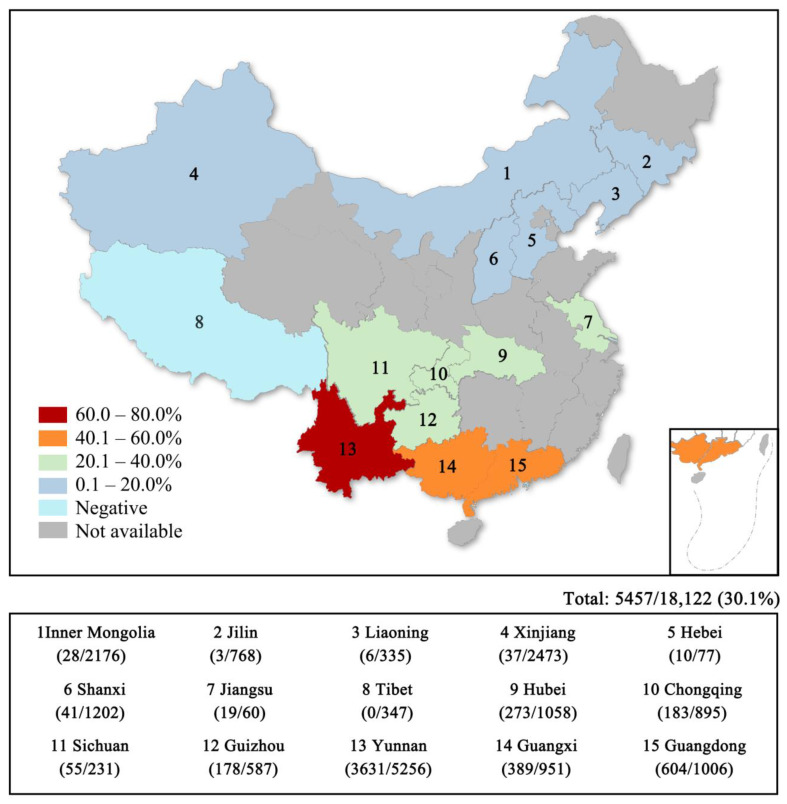
Seroprevalence of EHDV in tested domestic ruminants from 15 provinces in China between 2014 and 2019. Data from Duan et al. [21].

**Figure 3 vaccines-11-01123-f003:**
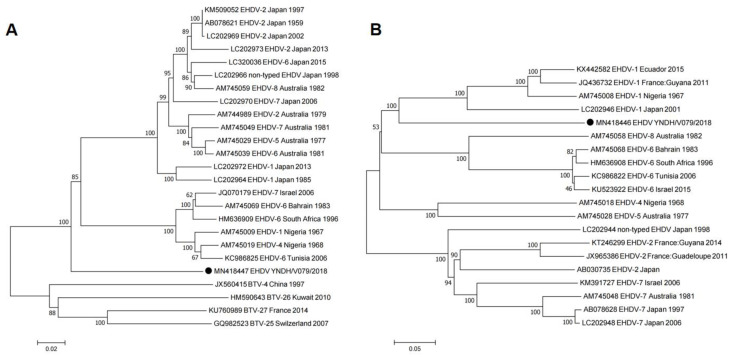
Phylogenetic analyses of EHDV based on Seg-3 (**A**) and Seg-2 (**B**) of YNDH/V079/2018 from Mangshi County, Yunnan Province, China (according to Yang et al. [29]).

## Data Availability

Not applicable.

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
