# Peer review of "Current Knowledge on Epizootic Haemorrhagic Disease in China"

_vaccines, 2023, doi:10.3390/vaccines11061123_

Round 1

Reviewer 1 Report

This review article is timely and relevant as EHDV is a worrying emerging epizootic virus. However, there are some aspects that, in my opinion, should be addressed.

- First, although is correct focusing the review on China´s situation, a more thorough general view of the whole world should be provided on the Introduction section. Aspects as, for example, where and when the last outbreaks of EHDV occurred, and which were the responsible serotypes.

- Also, as much importance is given to the possible reassortment of western with eastern topotypes, a more detailed explanation of the differences between those is required, what are the actual sequence differences between them, etc. Also relevant is to provide a summary of any reassortments reported to date, either naturally occurring, or produced in the laboratory: can all the serotypes reassort between them, are there any restriction or incompatibilities? The reason for this is that the authors claim that there is a putative reassortment event between a western and eastern topotype, but from the data presented this event is not clear enough: what is the real sequence difference between those? Can such differences only arrive by reassortment or are they likely to be generated by mutation? Might mutation explain the new described strain? Any explanation for this one? Why the reassortment is only happening with one of the circulating serotypes? Are any evidences of the western topotype circulating in any area of China or the closest countries? Are the putative reassortants more pathogenic than local strains?..

- Figure 1, a suggestion as it looks a bit strange as it is: the arrows pointing upwards and to the left might be combined in a single one, going from the cow, passing through the culicoides and going to the deer.

- In one sentence is said that there are 7 described serotypes of EHDV plus 2 recently reported, but then they are named EHDV1 to 10: this should be clarified.

- In the detection methods: the possible interference (or not) with other Orbiviruses should be commented for the immunological methods as it already is for PCR. On the other hand, no need for the detailed description of how a monoclonal (or the antibodies for ELISA)  was made is necessary for a review article.

- In the prevention section some comments on the possible development of a vaccine, taking in consideration the efforts made for other Orbiviruses is also advisable.  

Although the article is well written, some typos should be corrected. 

Author Response

Dear reviewer,

We appreciate you for your precious time in reviewing our manuscript and providing valuable comments. It was your valuable and insightful comments that led to improvements in the current version. We have carefully considered the comments and tried our best to address every one of them. We welcome further constructive comments if any. Below we provide the point-by-point responses.

Best regards.

Sincerely Jige Xin

22/5/2023

Reviewer 2 Report

The manuscript from Xin et al. reviewed advances on EHD/EHDV researches in China. Epidemic status of the disease in certain area inside China as well as methods for detection of the virus infection were summarized. Moreover, the authors discussed the risks and potential approaches for EHD prevention and control. The manuscript is in general well organized and written, with a few minor points should be stressed.

1. In Section 3 which summarized recent advances on detection methods of EHDV, I believe there are more methods developed such as LAMP which is rapid and sensitive.

2. In Fig 2, does blank represent no data available in these area/provinces? If so, this info could also be annotated together with other colours.

Author Response

Dear reviewer,

We appreciate you for your precious time in reviewing our manuscript and providing valuable comments. It was your valuable and insightful comments that led to improvements in the current version. We have carefully considered the comments and tried our best to address every one of them. We welcome further constructive comments if any. Below we provide the point-by-point responses.

  1. In Section 3 which summarized recent advances on detection methods of EHDV, I believe there are more methods developed such as LAMP which is rapid and sensitive.

Response 1: Thanks for your valuable reminder. The recent advances on detection methods were modified. New developments in technology of LAMP, as well as the liquid chip detection technique were supplied in Ln 245-263. The related information was also summarized in the Abstract section (Ln 30-31).

  1. In Fig 2, does blank represent no data available in these area/provinces? If so, this info could also be annotated together with other colours.

Response 2: Thank you for your insightful comment. The blank represents no data available in these provinces. They were annotated in gray color now and represent as Not available. (Ln 142)

Best regards.

Sincerely Jige Xin

22/5/2023

Round 2

Reviewer 1 Report

Although the authors have introduced additional data in the manuscript that makes it more understandable, there still are some parts that need some clarification.

That is the case, again, with the pretended western-origin reassortant virus. On lines 53-66 it is explained that segment 3 is very conserved, while segment 2 is the serotype-specific determinant. Thus, topotypes are determined essentially by segment 3 sequence. However, lines 143-150 explain that segment 2 from 7 strains isolated in China were assigned to western topotype, while their segment 3 was eastern. Again: what was the sequence divergence? Why that one could not arrive by mutation instead of reassortment? Is there any evidence for the circulation of any western topotype? This is a review article: the authors can also be critical about the results that are already published elsewhere.

Other major issue is the lack of information about the production/design of possible vaccines. The authors should at least mention how vaccination is done for other orbiviruses (type of used vaccines, some experimental, promising vaccines if any, etc.) and suggest possible vaccination strategies for this virus based on those.

On page 154 there is a new sentence: "and a similar situation also existed in the BTV reassorted strain [27] " Which BTV strain? Nothing at all is said about any BTV strain.

There still are some typos

Author Response

Dear reviewer,

Thank you for your insightful comment. We have carefully considered the comments and tried our best to address them.The point-by-point responses were provided, Please see the attachment,and the revision manuscript have been uploaded in the system.

Best regards.

Sincerely Jige Xin

Round 3

Reviewer 1 Report

I still find that the vaccination issue was not properly addressed.  The authors mention about several vaccine strategies (page 8, lines 313-319), but no references are given for those. Then, either reference the vaccination strategies for these viruses (EHDV), or explain what is being done for other Orbiviruses, different commercial and/or experimental vacines that are in development at the moment for any of them, etc. This is a review article, please elaborate a little bit more this section.

Some language editing is needed.

Author Response

Dear reviewer,

Thank you for your insightful comment. We have carefully considered the comment and tried our best to address it. The vaccination issue has been improved, including the references, the recent development and the prospect. The detailed responses were provided in the the attachment, and the revision manuscript has been uploaded in the system.

Best regards.

                                                                   Sincerely, Jige Xin
